# Risk Factors for Retained Hemothorax after Trauma: A 10-Years Monocentric Experience from First Level Trauma Center in Italy

**DOI:** 10.3390/jpm12101570

**Published:** 2022-09-23

**Authors:** Marta Rossmann, Michele Altomare, Isabella Pezzoli, Arianna Abruzzese, Andrea Spota, Marco Vettorello, Stefano Piero Bernardo Cioffi, Francesco Virdis, Roberto Bini, Osvaldo Chiara, Stefania Cimbanassi

**Affiliations:** 1General Surgery and Trauma Team, ASST GOM Niguarda, Piazza Ospedale Maggiore 3, 20162 Milan, Italy; 2Department of Surgical Sciences, Sapienza University of Rome, Piazzale Aldo Moro 5, 00185 Rome, Italy; 3Department of Pathophysiology and Transplants, State University of Milan, Via Festa del Perdono 7, 20122 Milan, Italy

**Keywords:** thoracic trauma, hemothorax, retained hemothorax, thoracic surgery, trauma, acute care surgery

## Abstract

Thoracic trauma occurs in 20–25% of all trauma patients worldwide and represents the third cause of trauma-related mortality. Retained hemothorax (RH) is defined as a residual hematic pleural effusion larger than 500 mL after 72 h of treatment with a thoracic tube. The aim of this study is to investigate risk factors for the development of RH in thoracic trauma and predictors of surgery. A retrospective, observational, monocentric study was conducted in a Trauma Hub Hospital in Milan, recording thoracic trauma from January 2011 to December 2020. Pre-hospital peripheric oxygen saturation (SpO_2_) was significantly lower in the RH group (94% vs. 97%, *p* = 0.018). Multivariable logistic regression analysis identified, as independent predictors of RH, sternum fracture (OR 7.96, 95% CI 1.16–54.79; *p* = 0.035), pre-admission desaturation (OR 0.96; 95% CI 0.77–0.96; *p* = 0.009) and the number of thoracic tube maintenance days (OR 1.22; 95% CI 1.09–1.37; *p* = 0.0005). The number of tubes placed and the 1° rib fracture were both significantly associated with the necessity of surgical treatment of RH (2 vs. 1, *p* = 0.004; 40% vs. 0%; *p* = 0.001). The risk of developing an RH in thoracic trauma should not be underestimated. Variables related to RH must be taken into account in order to schedule a proper follow-up after trauma.

## 1. Introduction

Thoracic trauma occurs in 20–25% of all trauma patients worldwide and represents the third cause of trauma-related mortality [1]. Most thoracic trauma is related to the blunt mechanism of injury. Penetrating mechanisms may still be involved [2]. One of the most common findings is acute hemothorax (HTX), often associated with rib fractures, pulmonary contusion, pneumothorax, and diaphragmatic injuries [3]. Hemothorax should be initially managed with a tube thoracostomy (TT) placement, but residual hemothorax is reported in up to 20% of patients [4].

A few authors define retained hemothorax (RH) as a residual hematic pleural effusion larger than 500 mL after 72 h of treatment with TT [5], and its incidence ranges between 17.6% and 28.9% [6,7]. It represents a major source of morbidity as it is associated with an increased risk of empyema, pneumonia, trapped lung, and prolonged hospital stay [8,9,10]. Firstly, the management of RH involves the replacement of TT or placement of an additional chest tube. Failure cases or selected patients may require surgical management such as Video-Assisted Thoracoscopic Surgery (VATS) or open thoracotomy within the first 2–7 days to improve outcomes [11].

However, definitions, diagnostic methods, and management indications vary from study to study, and a consensus is still lacking.

This study aims to investigate and identify risk factors for developing RH in thoracic trauma and predictors of surgery requirements.

## 2. Materials and Methods

Study design: This is a retrospective, observational, monocentric study conducted at Grande Ospedale Metropolitano Niguarda Hospital, Trauma Hub Hospital in Milan. Since 2011, all major trauma patients’ data were recorded prospectively in a trauma registry (Figure 1).

Population: All adult (>16 years old) trauma patients diagnosed with HTX between January 2011 and December 2020 were enrolled in the study. Patients with HTX were excluded if they were not treated by tube thoracostomy placement or treated only by thoracentesis. Dead or transferred patients within 72 h were also excluded. Retained hemothorax (RH) was defined as a residual clot estimated to be larger than 500 mL based on a CT scan of the chest that could not be drained by chest tube thoracostomy after 72 h of initial treatment [5]. The size of the hemothorax was quantified on CT scan images using Hazlinger’s method [12]. Our internal protocol for removing a chest tube after thoracic trauma is the following: Chest X-ray is performed 48h after every major thoracic trauma (ISS > 16 and or more than three rib fracture, and/or sternal fracture, and/or bilateral clavicle fracture and/or clavicle fracture + homolateral 1°/2° rib fracture, and/or scapula fracture). If an RH is evident at the chest X-ray, a CT scan is performed to quantify the HTX. In a traumatic patient, the placement of a thoracic tube is standardized in our experience. We place one or two thoracic tubes based on the presence of hemothorax and/or pneumothorax. The tubes are placed in the fifth intercostal space, one posterior and one anterior to the lung, to drain liquid and air. We do not choose different placements or strategies based on the CT findings.

Data source: Patients’ data were collected using Niguarda Trauma Registry. All patients’ medical records were reviewed to identify all interventions (TT placement and surgery). Patient demographics (age, sex, BMI, Charlson Comorbidity Index [13]), injury details, such as the mechanism of injury, Injury Severity Score (ISS), New Injury Severity Score (NISS) [14], and prehospital and admission physiology data were collected. Additionally, we recorded radiological features and volume of initial hemothorax, management of hemothorax, and retained hemothorax (tube thoracostomy placement and surgery). Associated thoracic injuries, ventilator days, overall hospital length of stay (LOS), mortality, and complications were also collected.

Outcomes: The primary aim of this study was to identify risk factors leading to the progression of HTX to RH. The secondary purpose was to determine the early predictors of surgical treatment requirements in RH.

Statistical analysis: Univariable and multivariable analyses were used to assess risk factors for RH and the need for surgery. Continuous, normally distributed variables were compared using the Kolmogorov–Smirnov test and Student’s *t*-test; continuous, non-normally distributed variables were compared with the Mann–Whitney U test. Categorical variables were compared using Fisher’s exact test or Pearson’s Chi-squared Test. All variables with a *p*-value of <0.05 on univariable analysis (and all clinically significant variables) were entered into a multivariable logistic regression analysis to identify independent risk factors for RH and the need for surgery. Data were reported as adjusted odds ratio (OR) with a 95% confidence interval (CI). Normally distributed data are expressed by the mean and standard deviation (SD); non-normally distributed data are described as the median and interquartile range (IQR). A *p*-value of <0.5 was considered statistically significant. Analyses were performed with MedCalc^®^ Statistical Software version 20.110 (MedCalc Software Ltd., Ostend, Belgium; https://www.medcalc.org (accessed on 3 March 2022).

## 3. Results

During the 10-year study period, a total of 2462 chest trauma presented to our Institution. One hundred thirty-seven patients had HTX, but only 93 met the study’s inclusion criteria. RH developed in 38 patients (40%), and 15 of them (39%) needed surgical treatment (VATS or open thoracotomy). All data are reported in Table 1. The majority of patients included in the study were male (86%), the median age was 48 (42–53), and the prevalent mechanism of injury was blunt trauma (82%). There was no significant difference in age, gender, and mechanism of injury between HTX and RH, while BMI was significantly higher in the RH group (27 ± 4 vs. 23 ± 2, *p* = 0.022). Forty-six percent of the population did not have a comorbidity (CCI 0), and only one patient in the RH group had CCI > 5, without a significant difference between the two groups. Three patients (8%) who developed RH used anticoagulant therapy. Pre-hospital peripheric oxygen saturation (SpO_2_) was significantly lower in the RH group (94% vs. 97%, *p* = 0.018). The pre-hospital physiology data did not show significant differences between the two groups. Patients who developed RH had a larger median hemothorax size on admission CT than those who did not develop RH, without a statistical significance (360 mL vs. 324; *p* = 0.140). Most HTX was managed with tube thoracostomy placement on ED admission (84%) or pre-admission (4%). Twenty-seven patients underwent TT placement after admission, including those needing an additional chest tube (five HTX and six RH). When compared to HTX, patients with RH more often required significantly longer TT duration days (6 vs. 12; *p* < 0.0001) and needed more TT placed (1 vs. 2; *p* = 0.030). The median chest tube size used was significantly larger in RH compared to HTX (32 Ch vs. 28 Ch; *p* = 0.024). First rib and sternum fractures were more often statistically associated with RH (16% vs. 4%, *p* = 0.041). RH patients had longer hospital LOS (30 days vs. 20 days, *p* = 0.066).

Multivariable logistic regression analysis (Table 2) identified, as independent predictors of RH, sternum fracture (OR 7.96, 95% CI 1.16–54.79; *p* = 0.035), pre-admission desaturation (OR 0.96; 95% CI 0.77–0.96; *p* = 0.009), and the number of TT maintenance days (OR 1.22; 95% CI 1.09–1.37; *p* = 0.0005). Analysis suggests that in the case of SpO_2_ < 90% a Relative Risk (RR) of development of RH of 1.16. Receiver operating characteristic (ROC) curves were constructed to determine cut-off values of continuous variables. As shown in Figure 2, when considering pre-admission SpO_2_, the optimal cut-off was 90% (sensitivity: 38.7%; specificity: 88.9%) with an area under the curve (AUC) of 0.659 (*p* = 0.015). In the case of total TT duration days, the RR of development of RH is 1.22. The AUC of ROC curve, as reported in Figure 3, is 0.818 (*p* < 0.0001), with a cut-off of 8 days (sensitivity: 78.8%; specificity: 74.4%).

A subgroup analysis was performed on RH patients to identify surgical treatment requirements’ predictors. About 40% of RH required further operative treatment (VATS or open thoracotomy). The majority of patients surgically treated were male (80%), with a median age of 54 (32–63) and blunt mechanism of injury (100%). The number of tubes placed was significantly associated with the necessity of surgical treatment (2 vs. 1; *p* = 0.004). Among associated thoracic injuries, first rib fracture is the only variable significantly related to surgical treatment of RH (40% vs. 0%; *p* = 0.001). The mean time to surgery was 6.7 days (3.3–9.7).

## 4. Discussion

The literature reports several risk factors of retained hemothorax: high chest AIS score, low hematocrit rate, initial HTX volume (with an increased risk of 15% every 100 mL/day of HTX), low GCS, IOT at admission, number of TT placed, and length of time with thoracostomy [6,7,15]. In this study, 41% of patients with HTX on admission developed RH. Pre-admission SpO_2_ < 90%, sternum fracture, and TT duration > eight days were identified as independent factors associated with the development of RH. The mortality rate was 6.4%. The median size of HTX was 353 mL (233–397), and, as described by Prakash et al., the development of an RH increased in moderate and large initial HTX with 44.3% and 39.1% RH risk rates, respectively [7]. To our knowledge, pre-admission physiology data were never investigated so far, and pre-hospital SpO_2_ level < 90% seems to be related to increased RH risk. Sternum fracture increases the odds of developing RH seven-fold as a 1° rib fracture appears to be related to needing a surgical approach. Those injuries frequently result from high-energy blunt trauma with anteroposterior chest compression. They are associated with pulmonary contusion [16,17,18]. In other studies, the duration of tube thoracostomy > 8 days was found to be an associated factor. Villegas et al. reported a median of 5 days TT duration in RH patients. TT is likely left in place longer to drain residual HTX that is already clotted although still visible on X-rays [15,19,20]. Regarding the surgical treatment of RH, several studies suggest VATS or open thoracotomy within the first 2–7 days rather than additional TT placement to improve outcomes [21,22]. To our knowledge, no study analyzed predictors for surgery requirements. The reported rate of RH requiring definitive surgical treatment varies between 28.3 and 54.7% [6,7,15]. This study confirms that 15 (39%) of RH patients needed surgical intervention (VATS or thoracotomy). The red tag on triage, first rib fracture, and the number of TTs placed significantly correlate with surgical treatment. More recent guidelines showed that surgical interventions should be preferred over additional TT placement because of the high risk of failure of the latter. Our findings and experience are in line with this statement [3]. Moreover, in the last practice guidelines for the management of RH published in the American Journal of Surgery [23], the authors recommend early VATS more than thrombolytic therapy in line with our daily approach and results. On the other hand, we do not routinely use a pig-tail catheter in patients with simple hemothorax after trauma for the risk of failure related to the size of the catheter. In fact, in our analysis the size of catheter seems to be related to the risk of failure of a thoracic drain placement. Moreover, even if the length of stay of a thoracic tube could be considered not a predictor but a consequence of the RH, in our experience it also could be used as a red flag to not underestimate the risk of the need for surgical intervention. Regarding the mean time to surgery reported in our series, we seem to be outside the comfort zone reported in the literature of 4 days [24]. This could be related to the absence of a standardized protocol for retained hemothorax in our department. On the other hand, VATS has been the first choice during the last few decades in order to reduce comorbidity related to an open thoracotomy. This approach is in line with the most recent shreds of evidence in the literature.

This study has several limitations. The small sample size represents the main problem related to identifying significant predictors, in particular regarding the reliability of the multivariate analysis performed. The retrospective nature is another considerable limitation considering the heterogeneity of treatments and protocols adopted during the study period (10 years). Finally, the monocentric nature should be considered, and our results should be validated with further multicentric studies conducted on a larger population.

## 5. Conclusions

RH is an essential source of morbidity and mortality in thoracic trauma patients. There is still a lack of a consensus on its definition, and the literature described multiple developing risk factors. In our study, pre-admission SpO_2_ < 90%, sternum fracture, and TT duration > eight days seem to be the most interesting predictors of RH. This highlights the need for studies based on univocal diagnostic and therapeutic protocols to delineate better those patients who would benefit from preventive strategies. Further studies on large samples are required to identify early predictors of surgery in RH to optimize the timing and strategy of operative intervention and improve patients’ outcomes.

## Figures and Tables

**Figure 1 jpm-12-01570-f001:**
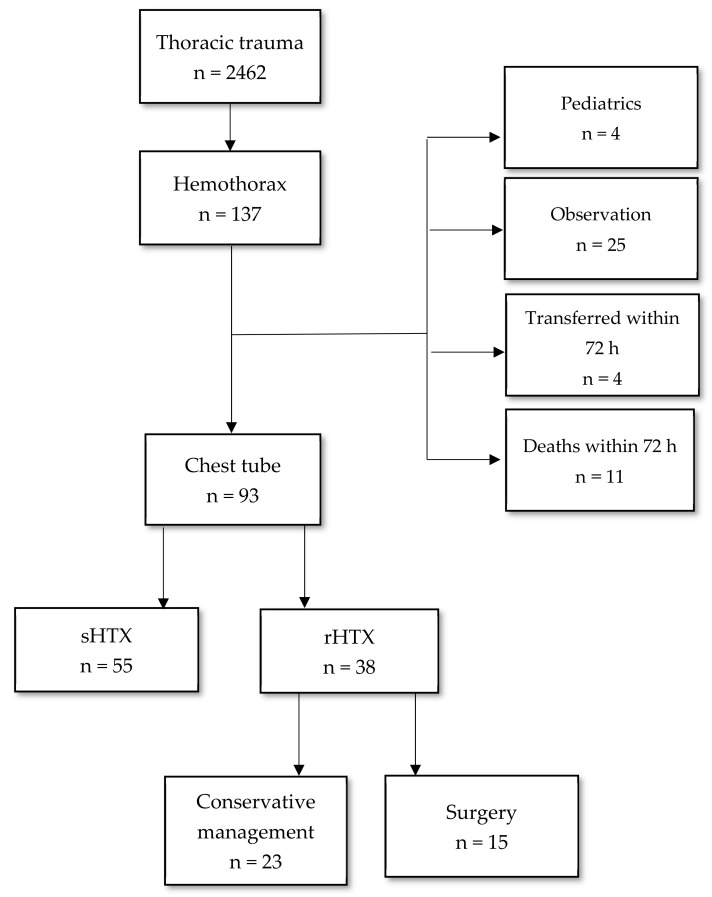
Study flow-chart—sHTX (simple hemothorax); rHTX (retained hemothorax).

**Figure 2 jpm-12-01570-f002:**
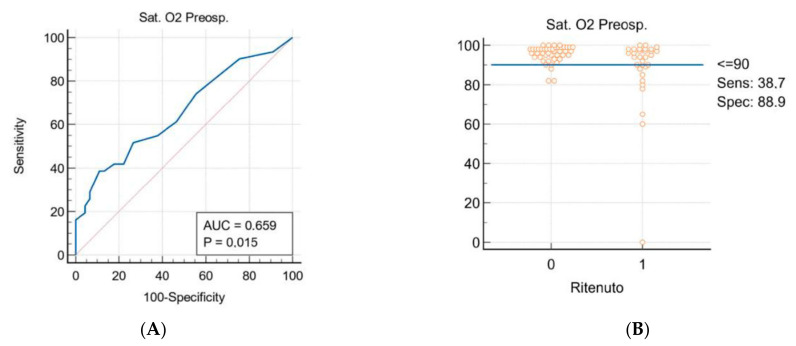
(**A**) SpO_2_ ROC Curve; (**B**) patients’ distribution based on pre-admission SpO_2_.

**Figure 3 jpm-12-01570-f003:**
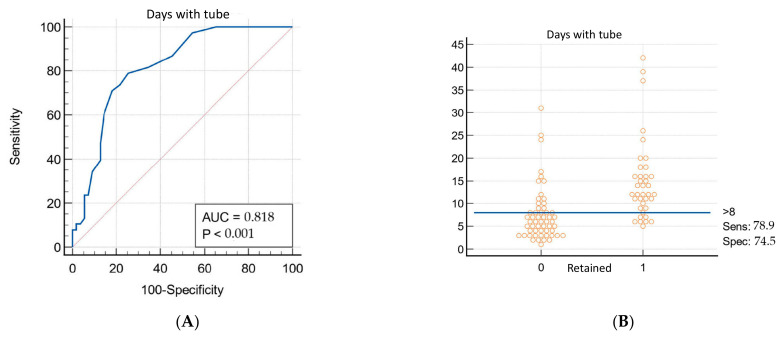
(**A**) No TT duration days ROC curve; (**B**) patients’ distribution based on No TT duration days.

**Table 1 jpm-12-01570-t001:** Demographic data and univariate analysis of predictive factor for RH. TT: thoracic tube.

	HTX *n* = 55	rHTX *n* = 38	Tot. *n* = 93	*p* Value
**Demographic data**				
Age, yrs., median (IQR)	47 (40–53)	49 (36–56)	48 (42–53)	0.698
Gender, male, *n* (%)	46 (84)	34 (90)	80 (86)	0.427
BMI, Kg/m^2^, mean (±SD)	23 (±2)	27 (±4)	23 (±3)	**0.** **022 ***
**Pre-hospital data**				
GCS, median (IQR)	15 (14–15)	15 (15–15)	15 (15–15)	0.537
Heart Rate, beats/min, median (IQR)	100 (91–110)	100 (90–105)	100 (98–104)	0.670
Respiratory Rate, median (IQR)	18 (16–18)	18 (16–21)	18 (16–19)	0.353
Systolic Blood Pressure, mmHg, median (IQR)	112 (±33)	108 (±35)	110 (±33)	0.559
Peripheric oxygen saturation (SpO_2_), %, median (IQR)	97 (95–98)	94 (90–97)	96 (95–97)	**0.** **018 ***
**In-hospital data**				
Heart rate, bpm, median (IQR)	100 (94–111)	92 (87–105)	100 (92–110)	0.357
Systolic Blood Pressure, mmHg, mean ± SD	113 (±34)	121 (±35)	117 (±35)	0.289
SpO_2_, %, median (IQR)	99 (98–100)	98 (69–99)	98 (98–99)	0.136
INR, median (IQR)	1.1 (1.1–1.2)	1.2 (1.1–1.2)	1.2 (1.1–1.2)	0.298
**Thoracic drain management**				
TT placed pre-hospital, *n* (%)	2 (4)	2 (5)	4 (4)	0.705
TT placed on ED admission, *n* (%)	45 (82)	33 (87)	78 (84)	0.519
TT placed inpatient ward, *n* (%)	16 (29)	11 (29)	27 (29)	0.988
Total TT days, median (IQR)	6 (5–7)	12 (11–15)	1 (1–2)	**<0.0001 ***
Number of TT placed, median (IQR)	1 (1–2)	2 (1–2)	1 (1–2)	**0.** **030 ***
Chest tube size, Ch, median (IQR)	28 (28–29)	32 (28–32)	28 (28–32)	**0.** **024 ***
**Associated thoracic injuries**				
Pneumothorax, *n* (%)	39 (71)	29 (76)	68 (73)	0.565
Pulmonary contusion, *n* (%)	33 (60)	25 (66)	58 (62)	0.573
First rib fracture, *n* (%)	2 (4)	6 (16)	8 (9)	**0.** **041 ***
N° rib fractures, median (IQR)	6 (5–7)	9 (5–11)	6 (6–9)	0.070
Sternum fracture, *n* (%)	2 (4)	6 (16)	8 (9)	**0.** **041 ***
Clavicle fracture, *n* (%)	11 (20)	3 (8)	14 (15)	0.110
Scapula fracture, *n* (%)	13 (24)	2 (5)	15 (16)	**0.** **018 ***

* *p* < 0.05.

**Table 2 jpm-12-01570-t002:** Multivariate logistic regression analysis.

	Odds Ratio	CI 95%	*p*-Value
Sternum fracture	7.96	1.16–54.79	**0.035 ***
Pre-hospital SpO_2_	0.86	0.77–0.96	**0.009 ***
Total TT days	1.22	1.09–1.37	**0.0005 ***

* *p* < 0.05.

## Data Availability

The data presented in this study are available on request from the corresponding author. The data are not publicly available due to the restrictions (e.g., privacy or ethical).

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
