# Peer review of "Risk Factors for Retained Hemothorax after Trauma: A 10-Years Monocentric Experience from First Level Trauma Center in Italy"

_jpm, 2022, doi:10.3390/jpm12101570_

Round 1

Reviewer 1 Report

Thank you for the opportunity to review this manuscript.

I have the following questions.

Abstract.

Please define TT

Please clarify the “tubes”, mean “thoracostomy” or “chest tubes”

What does it mean TT maintenance days?

Methods

Please define adult patients.

Please clarify what do you mean by stating that RH could not be trained after 72 hours.

What is your protocol to remove chest tubes? What are indications for a placement of the second TT?

How do you  decide to proceed to VATS vs thoracotomy?

Please do subsections: study design, population, data source, outcomes, statistical analysis

The following statement is the goal of your study but not the outcome. The outcome here is a risk of RH

“The primary outcome of this study was to identify risk factors leading to a 74 progression of HTX to RH.”

This is the second goal, but outcome

“The secondary outcome was to identify the early predictors of 75 surgical treatment requirements in RH.”

Results

Please provide only information relative to your goals.

Avoid repeating the same information in tables and the text.

“Results” section is too long and difficult to follow.

There is no clinical difference between these values. Please provide more information to justify such conclusion. in order to make it clinically relevant, I would report a proportion of patients in both groups with and without RH who had less than 90% oxygen saturation.

 “Pre-hospital peripheric oxygen saturation (SpO2) was significantly lower in the RH group (94% vs. 97%, p=0,018)” 

This is a misleading statement as there is no statistically significant here.

 “Patients who developed RH had larger median haemothorax size on admission 110 CT than those who did not (360 mL vs. 324; p = 0,140).”

What is the goal of reporting this information? Do you use EFAST results to predict RH?

“In hemodynamically stable patients, we usually perform E-FAST and chest X-Ray at admission to Emergency Department (ED). About 55% of RH had positive E-FAST on admission. Moreover, chest X-Ray was performed in 82% of these patients (82% vs 62%; p 109 = 0,042??).”

RH patients required prolong TT. This is obvious. RH was not developed because of prolong TT, which is  a treatment.

How do you explain sternal fracture as a predictor of RH?

Tables.

Make sure all tables and figure have clear captions.

Please explain this variable “No TT placed, median (IQR)”

Discussion

This is too short. Does not provide a comprehensive review of the existing literature on the topic, and doesn’t clearly explain the novelty of the study.

Author Response

1.1 Please define TT … Please clarify the “tubes”, mean “thoracostomy” or “chest tubes” … What does it mean TT maintenance days?

TT means thoracic tube, and the maintenance days are the number of days during which the TT was in place.  We have changed the abstract as requested, and we hope it is clearer now. 

1.2 Please define adult patients.

              Thank you for this precise suggestion. Adult patients in our series are those older than 16.

1.3 Please clarify what do you mean by stating that RH could not be trained after 72 hours.

Following the recent literature, we have used this statement to classify one haemothorax as retained. If, 72 hours after placing a thoracic tube, more than 500 ml of haemothorax were found at the imaging performed, we have classified the HTX as an RH.

1.4 What is your protocol to remove chest tubes? What are indications for a placement of the second TT? How do you decide to proceed to VATS vs. thoracotomy?

We want to thank Reviewer 1 for this accurate suggestion which allowed us to add to the paper our protocol for removing a chest tube in thoracic trauma. We have added this statement on lines 65-69, and we hope that the M&M section is now more clear. The placement of a second thoracic tube depends on the CT-scan findings. Regarding the VATS vs. Thoracotomy Q, we have performed VATS in almost all cases during the last decade, leaving the thoracotomy only for VATS failures.

1.5 Please do subsections: study design, population, data source, outcomes, statistical analysis.  

As requested, this enclosed R1 version has the subsections in the M&M section.

1.6 Avoid repeating the same information in tables and the text … “Results” section is too long and difficult to follow.

As requested, we have changed the results section and tables. We hope this new R1 version is clearer now than the previous one.

1.7 The following statement is the goal of your study but not the outcome. The outcome here is a risk of RH “The primary outcome of this study was to identify risk factors leading to a 74 progression of HTX to RH.” This is the second goal, but the outcome “The secondary outcome was to identify the early predictors of 75 surgical treatment requirements in RH.”

Thank you for this precise clarification. We have changed the word “outcome” with “aim” in the paper to clarify the issue raised.

1.8 There is no clinical difference between these values. Please provide more information to justify such conclusion. in order to make it clinically relevant, I would report a proportion of patients in both groups with and without RH who had less than 90% oxygen saturation. “Pre-hospital peripheric oxygen saturation (SpO2) was significantly lower in the RH group (94% vs. 97%, p=0,018)” 

Thank you for this clarification request. We have reported the differences in pre-hospital SpO2 to underline the subsequent result of the multivariable logistic regression. Even if we know that it seems to be no clinical differences between those two values, in our series, this data is valuable to underline how pre-hospital saturation should be considered a possible red flag for the incidence of RH during the hospital stay.

1.9 This is a misleading statement as there is no statistically significant here. “Patients who developed RH had larger median haemothorax size on admission 110 CT than those who did not (360 mL vs. 324; p = 0,140).”

We agree with R1 and have changed the statement clarifying the absence of statistical significance.

1.10 What is the goal of reporting this information? Do you use EFAST results to predict RH? “In hemodynamically stable patients, we usually perform E-FAST and chest X-Ray at admission to Emergency Department (ED). About 55% of RH had positive E-FAST on admission. Moreover, chest X-Ray was performed in 82% of these patients (82% vs 62%; p 109 = 0,042??).”

We agree entirely with R1 regarding this suggestion and have removed the statement from the text.

1.11 RH patients required prolong TT. This is obvious. RH was not developed because of prolong TT, which is  a treatment. … How do you explain sternal fracture as a predictor of RH?

We agree with the reviewer's suggestion. A long necessity of a TT is not a prognostic factor related to the RH but a red flag to understand that those patients could require a more aggressive surgical approach. Regarding the second question, our hypothesis relates to the well-known relationship between sternal fracture and major traumatic injuries (ISS> 16). We believe that, as pre-hospital desaturation, a sternal fracture could be a red flag not to underestimate the risk of RH.

1.12  Make sure all tables and figures have clear captions … Please explain this variable “No TT placed, median (IQR).”

We have reviewed all figures and tables. We have deleted the redundant ones and changed some captions in the test. We hope that this new version of the manuscript is more precise. “No TT placed” is the number of thoracic tubes placed during the hospital stay.

1.13 This is too short. Does not provide a comprehensive review of the existing literature on the topic, and doesn’t clearly explain the novelty of the study.

We want to thank Reviewer 1 for this suggestion which allowed us to ameliorate our paper. We have completed the discussion section, adding relevant articles related to our findings.

Reviewer 2 Report

Thank you for the opportunity to review the paper entitled “Risk factors for retained haemothorax after trauma: 10-years monocentric experience from First Level Trauma Center in Italy.” By Rossmann and colleagues. 

I enjoyed reading this article, I do have some comments however.

Line 2: the second part of the title is grammatically incorrect, try “A 10-years monocentric experience from a First Level Trauma Center in Italy”

The first box of the flowchart has the number of enlisted patients partially erased, please upload a correct picture. 

This is an interesting study however it does not add any significant information to known trauma data. Rib and sternal fractures increase the likelihood of retained hemothorax which correlates to the dynamics and mechanics of the initial trauma, not a surprising finding here.

Additionally, the authors mention some patients being taken to surgery through an open thoracotomy vs VATS, based on what criteria did they choose one over the other, please clarify. I am very interested to know the location of chest tube placement as this is also an important factor in whether hemothoraces retain or not. If the authors can provide this data, it would make the paper a lot stronger. 

Altogether a well-structured paper, however the conclusions are not very useful and do not provide any new or clinically relevant data. 

Author Response

2.1 Line 2: the second part of the title is grammatically incorrect, try “A 10-years monocentric experience from a First Level Trauma Center in Italy”

              We want to thank R2 for this correction. We have changed the title as suggested.

2.2 The first box of the flowchart has the number of enlisted patients partially erased, please upload a correct picture. 

              We have changed the picture, and now it should be more precise.

2.3 This is an interesting study however, it does not add any significant information to known trauma data. Rib and sternal fractures increase the likelihood of retained hemothorax which correlates to the dynamics and mechanics of the initial trauma, not a surprising finding here.

We agree with R2. On the other hand, we believe that it could be important to underline that red flags should be taken into account in patients with thoracic trauma to understand when it is important to perform VATS because the possibility of failure with a thoracic tube is high.

2.4 Additionally, the authors mention some patients being taken to surgery through an open thoracotomy vs. VATS, based on what criteria did they choose one over the other, please clarify. I am very interested to know the location of chest tube placement as this is also an important factor in whether hemothoraces retain or not. If the authors can provide this data, it would make the paper a lot stronger. 

              We want to thank R2 for these suggestions. The choice of VATS vs. open thoracotomy is related only to the surgeon's expertise. During the last five years, we have performed VATS routinely, but in the early part of our experience, we used to perform also thoracotomy in complicated cases. Regarding the TT placement issue raised, in a traumatic patient, the placement of a thoracic tube is standardized in our experience. We place one or two thoracic tubes based on the presence of hemothorax and/or pneumothorax. The tubes are placed in the fifth intercostal space, one posterior and one anterior to the lung, to drain liquid and air. We do not choose different placements or strategies based on the CT findings. We have clarified this in the materials and methods section.

Thanks to a mother tongue colleague, we extensively edited the English language and style to satisfy the minor English revisions requested both from Reviewer 1 and Reviewer 2.

We declare that the article is original and was never presented and/or submitted elsewhere, and all Authors approve the revised version.

Round 2

Reviewer 1 Report

1.In the whole manuscript please  "tubes" (like in the abstract) to either "chest tubes" or  "thoracostomy tube". 

2.please add to the abstract the proportion of patients who developed retained hemothorax

3.Under "outcomes" please only state your primary outcome, which is retained.  hemothorax. There is no need to state your aims here again as you have stated it already in the Introduction.

4.please remove SpO2 94vs97% from the abstract and instead report it as you do in the results: "Analysis suggests that in the case of SpO2 < 90% , a Relative Risk (RR) of development RH of 1,16."

Reviewer 2 Report

Thank you for answering my comments, I have no further suggestions